# Sleep quality and associated factors among adult hypertensive patients attending a chronic follow up care clinic in northwest Amhara regional state referral hospitals, Northwest Ethiopia

Teshale Ayanaw[1], Mahlet Temesgen[2], Abere Woretaw Azagew[1], Yohannes Mulu Ferede[1]*

1 Department of Medical Nursing, School of Nursing, College of Medicine and Health Sciences, University of Gondar, Gondar, Ethiopia, 2 Department of Surgical Nursing, School of Nursing, College of Medicine and Health Sciences, University of Gondar, Gondar, Ethiopia

* yohannesmulu50@gmail.com

**Data Availability Statement:** All relevant data are within the paper and its Supporting Information files.

## Abstract

### Introduction

Sleep plays an essential role in every individual's life since it contributes greatly to their well-being and health. Poor sleep quality is one of the most common consequences of hypertension (HTN), which is a leading public health problem. In Ethiopia, it is the main health concern among hypertension patients. Thus, in the study area, there is limited information regarding the investigation of sleep quality among hypertension patients. Therefore, this study aimed to assess sleep quality and associated factors among adult hypertensive patients attending a chronic follow up care clinic in northwest Amhara regional state referral hospitals, Northwest Ethiopia.

### Methods

An institution-based cross-sectional study was conducted among adult HTN patients from March 15/2021 to May 15/2021 in Northwest Amhara Regional State Referral Hospitals. A systematic random sampling method was used to select the study participants. A face-to-face interviewer-administered questionnaire with chart review was used. The Pittsburgh Sleep Quality Index (PSQI) for assessing sleep quality was used in this study. A binary logistic regression was used to see the association between dependent and independent variables. Variables having a p-value < 0.05 with a 95% CI were considered statistically significant in the multivariable logistic regression analysis.

### Results

In this study, 563 hypertensive patients were included, with a response rate of 96.9%. The overall prevalence of poor sleep quality was 37.7% (95% CI: 33.4%–41.6%). Female gender

**Funding:** The University of Gondar sponsored the study; however, the funders had no role in study design, data collection and analysis, decision to publish, or preparation of the manuscript.

**Competing interests:** The authors have declared that no competing interests exist.

**Abbreviations:** AOR, Adjusted Odd Ratio; BMI, Body Mass Index; HADS, hospital anxiety and depression scale; HTN, Hypertension; NREM, non-rapid eye movement; OSA, obstructive sleep apnea; PSQI, Pittsburgh sleep quality index.

(AOR = 2.55, 95% CI: 1.55–4.18), age ≥ 65 years (AOR = 4.07, 95% CI: 2.07–7.97), over-weight (BMI) ≥ 25 kg/m2 (AOR = 1.68, 95% CI: 1.06–2.65), WHO stage II hypertension (AOR = 1.78, 95% CI:1.01–3.12), poor physical activity (AOR = 2.39, 95% CI:1.41–4.05), participants who had depression (AOR = 2.03,95% CI:1.24–3.34) and participants who had anxiety (AOR = 1.89, 95% CI: 1.16–3.03) were factors associated with poor sleep quality.

## Conclusions

In this study, more than one-third of the study participants had poor sleep quality. Female gender, age ≥ 65 years, overweight (BMI) ≥ 25 kg/m2, stage II hypertension, physical inactivity, depression, and anxiety were all associated with poor sleep quality. Managing depression and anxiety disorders, encouraging physical activity, and providing health education about weight loss are all necessary for HTN patients.

## Introduction

Hypertension, also known as high blood pressure, is characterized by elevated blood pressure [1]. The global prevalence of hypertension was estimated to be 1.13 billion people, with the majority living in low and middle-income countries [2]. The number of hypertension cases in Africa has been steadily increasing, and by 2030 it is expected to reach 216.8 million [3]. By 2025, there will be more than 125 million hypertensive people in Sub-Saharan Africa [4]. In Ethiopia, the prevalence of hypertension ranged from 7% to 37% [5].

Sleep plays an essential role in everyone's life; it allows people to relax, refresh their bodies, minds, and emotions, and it improves general health [6]. Overall, sleep is as important as nutrition or exercise in maintaining overall health [7, 8]. Globally, the prevalence of poor sleep quality among hypertensive patients varies greatly, with a range of 14.9% to 85.7% [9]. In Africa, the prevalence of poor sleep quality among adult hypertensive patients was 44% [10]. In Ethiopia, the prevalence of poor sleep quality among adult hypertensive patients was 35.5% [11].

Different evidence explored that physical inactivity, diastolic blood pressure (Stage I & II HTN), cigarette smoking, alcohol consumption, systolic blood pressure (Stage II HTN), age, gender, obesity, patients with comorbid diseases, duration of medication use, years of diag-nosed hypertension, occupation, marital status, living in rural areas, depression, and anxiety have all been linked to poor sleep quality in HTN patients [9, 11–18].

Poor sleep quality is associated with many negative health outcomes and an increased risk of mortality. Insomnia, obstructive sleep apnea, restless legs syndrome, impaired cognitive per-formance, stroke, depression, decreased productivity, decreased job performance, more prone to accidents, decreased quality of life, increased health care utilization, and a high rate of psy-chiatric comorbidities were among the negative health consequences [19–21].

Although blood cells had an effect on obstructive sleep apnea syndrome (OSAS), white blood cell count (WBC) to mean platelet volume (MPV) ratio (WMR) had a greater impact on OSAS than neutrophil-lymphocyte ratio (NLR) and platelet-lymphocyte ratio (PLR) [22]. Likewise, OSAS was associated with a higher risk of neurovegetative diseases as well as poor cognitive function. Almost seventy studies indicate that CPAP (continuous positive airway pressure) therapy improves neurocognitive performance in OSA patients [23].

Consequently, preventing and reducing morbidity and mortality from non-communicable diseases (NCDs), including hypertension, is the target of the World Health Organization (WHO) at 25% by the end of 2025, and tackling poor sleep quality has been identified as a key measure in achieving this target [24]. In Ethiopia, the occurrence of hypertension has increased from time to time [25]. Related to this, most of the HTN patients were vulnerable to a variety of health-related problems, of which poor sleep quality is the major one. Concurrent occurrences of poor sleep quality with HTN make clinical management more complicated, so understanding the magnitude and major factors of poor sleep quality plays a key role in identifying and treating those problems early.

In Ethiopia, there is limited information about the study of sleep quality among hypertension patients. Hence, conducting this study was able to give evidence on the prevalence as well as major factors of poor sleep quality. In addition, the results of this study provide information for health care providers and policymakers to pay attention to adult HTN patients to improve their quality of life through improving service quality, mainly on sleep, and also serve as baseline information for further study. Therefore, this study aimed to assess poor sleep quality and associated factors among adult hypertensive patients attending chronic follow-up care clinics at northwest Amhara regional state referral hospitals in northwest Ethiopia.

## Methods and materials

### Study design, period, and setting

An institution-based cross-sectional study was conducted from March 15, 2021, to May 15, 2021, among adult hypertensive patients attending a chronic follow-up care clinic in Amhara Regional State Referral Hospitals. The Amhara Regional State has eight referral hospitals: Debre Birhan, Debere Markos, Dessie, Wolldya, Felege Hiwot, Tibebe Gion, Debretabor, and the University of Gondar Comprehensive Referral Hospital. Despite this, in Northwest Amhara Regional State, there are five hospitals. Of these, three hospitals were randomly selected by using the lottery method. These hospitals are Debretabor, Tibebe Gion, and the University of Gondar Comprehensive Referral Hospital. There are 2116 adult HTN patients in the study areas. Based on January 1, 2021, Ethiopia Demography and Health organization statistics (EDHS), Amhara Region has a total population of 30,087,766 with an equal number of females and males [26].

### Source and study population

All adult HTN patients who had a chronic follow-up in the three selected hospitals in Northwest Amhara regional state were the source population. All adult HTN patients who were attending a chronic follow-up care clinic from March 15/2021 to May 15/2021 in the selected hospitals were included in the study population.

### Inclusion and exclusion criteria

All HTN patients aged ≥18 years old who were attending a chronic follow-up care clinic during the data collection period were included in the study. Adult HTN patients who were seriously ill were excluded from the study.

### Sample size determination and sampling technique

The sample size was determined by using a single population proportion formula considering a 95% confidence interval (CI), a 35.5% population proportion of the previous study [11], and

a 5% margin of error.

$$n = \left(Z\frac{\alpha}{2}\right)2 * p\,(1-p)/\,(d)2$$

n = the initial sample size

Z α/2 = Standardized normal distribution value for the 95% CI, = 1.96

P = proportion of poor sleep quality (35.5%)

d = margin of error 5%

N = $(1.96)^2 \times 0.355\,(1-0.355)/(0.05)^2$ = 352

Design effects were used (1.5 x 352 = 528) and by adding a 10% non-response rate, the final sample size was 581. Similarly, the sample size was calculated by using Epi-info version 7. Statistically significant variables were selected from the previous study to calculate the sample size, and among the calculated samples, 128 was found to be the largest sample size. Hereafter, among the two calculated samples, the sample size of the first objective (n = 581) was larger than the second objective (n = 128), so the largest sample size was taken as the final sampling size (n = 581). Among the total of five hospitals found in the Northwest Amhara region, three of them were selected by using the lottery method. And then samples were proportionally allocated to each selected hospital. Following that, the desired sample size was selected using a systematic random sampling technique (every four HTN patients were selected).

## Operational definitions

**Poor sleep quality.**   If the global score on PSQI is > 5, patients will be classified as having poor sleep quality, otherwise not [27].

**Duration of sleep.**   A person who sleeps ≤ 6 hours per 24 hours is considered to have short sleep, whereas a person who sleeps ≥ 9 hours per 24 hours is considered to have long sleep [28].

**Subjective sleep quality.**   It is the combination of parameters such as sleep initiation, sleep continuity (number of awakenings), and depth of sleep [29].

**Sleep latency.**   It has been defined as the time it takes from turning off the lights to the start of the first epoch of any stage of sleep [30].

**Body mass index (BMI).**   A person's weight was classified as underweight (BMI< 18.5), normal body weight (BMI 18.5–24.9), overweight (BMI 25–29.9), and obese (BMI ≥30) [31].

**Physical activity.**   It is defined as a person who reports regular aerobic exercise (e.g., walking, jogging, etc.) for at least 30 minutes per day, five days per week, or whose occupation requires daily physical exertion [32].

**Alcohol consumption.**   It has been defined as more than one drink per day, more than seven drinks per week, or more than three drinks with heavier alcohol use such as beer, wine, and other hard liquor in the previous 30 days [33].

**Cigarette smoking.**   never smoker; if patients had no history or were not currently smoking a cigarette, former smoker; if patients had a history of smoking habit, but he or she stopped smoking currently, current smoker; if patients knew they were currently smoking a cigarette at least once within the last 30 days [34].

**Chewing Khat.**   If the client had consumed Khat within the past month, he or she was classified as a current chewer; if he or she had a history of chewing, he or she was classified as a former chewer; otherwise, he or she was classified as a never chewer [35].

**Depression.**   From a total of depression-related questions, if patients had a total score of > 8, they were considered to have depression, and if they had a score of 8, they were considered to have no depression [36].

**Anxiety.**   From a total of anxiety-related questions, if the patients have a total score of > 8, they are considered to have anxiety, and patients having a total score of ≤ 8 are defined as having no anxiety [36].

## Data collection tools and procedures

A face-to-face interview administered questionnaire with a chart review was used to collect the data. The questioners include socio-demographic-related factors, clinically related factors, behavioral-related factors, and psychosocial-related factors. A validated PSQI questionnaire tool with a Cronbach alpha of 0.77 was used [11, 37]. Sleep quality was divided into seven components, including subjective sleep quality, sleep latency, sleep duration, habitual sleep efficiency, sleep disturbances, use of sleeping medications, and daytime dysfunction in the previous month. To measure sleep quality, a global score called the PSQI was used. Each of the seven components has a score ranging from 0 to 3, resulting in a total score ranging from 0 to 21. The higher the score, the worse the quality of sleep, so scores > 5 indicate poor sleep quality [35]. For assessing anxiety and depression, the hospital anxiety and depression scale (HADS) was used. It was originally developed by Sigmund and Snaith [38]. The anxiety-related question contained 7 Likert scales; if the patients had a total score of > 8, they were considered to have anxiety, and patients having a total score of ≤ 8 were said to have no anxiety [36]. Depression-related questions had seven Likert scales; if patients had a total score of > 8, they were considered to have depression; if patients had a score of 8, they were considered to have no depression [36].

## Data quality management

A pretest was conducted on 5% of the total sample size (29 adult HTN patients) at Koladiba primary hospital's chronic follow-up care clinic before the actual data was collected. Four BSc nurses and one MSc medical nurse participated in the data collection process. Half-day training was given to data collectors and supervisors by the principal investigator about the objective of the study, data collection procedures, and ethical issues. The collected data was checked for completeness and errors were corrected by the principal investigator daily. Also, the internal reliability of the tool was assessed by Cronbach's alpha, which was 0.77, 0.80, and 0.86 for PSQI, anxiety, and depression, respectively.

## Data processing and analysis

The data was coded, cleaned, and entered into EPI Data Version 4.6 before being exported to SPSS Version 23.0 for analysis. The model fitness was checked by using the Hosmer-Lemeshow goodness of fit test with a p-value of 0.80 and interpreted as a model fit. Also, the multicollinearity assumptions were checked by using the variance inflation factor (VIF). Descriptive statistics like tables and figures were used for data presentation. Both bivariable and multivariable logistic regression were used to identify variables associated with the outcome variable (sleep quality). The degree of association was interpreted by using crude odds ratios (COR) and adjusted odds ratios (AOR) with 95% confidence intervals, and a P-value less than 0.05 was considered statistically significant. Variables having a P-value of less than 0.25 in the bivariable logistic regression analysis were entered into multivariable logistic regression. In the multivariable logistic regression analysis, variables having a P-value of less than 0.05 were considered variables statistically associated with sleep quality.

### Ethical consideration

Ethical clearance was obtained from the institutional ethical review committee of the school of nursing and the college of health science of the University of Gondar (Ref.No: S/N172/7/2013). In addition, permission and a supportive letter were obtained from the hospital's clinical directors. All participants were informed about the objective of the study. They were also informed about their full right to not participate in or withdraw from the study at any time. Written informed consent was obtained from each study participant.

## Results

### Socio-demographic characteristics of adult HTN patients'

In this study, 563 adult HTN patients participated, with a response rate of 96.9%. The median (IQR) age of the participants was 65 (45–85 years old). More than half 331 (58.8%) were females and 347 (61.6%) were married. Among study participants, 215 (38.3%) were unable to read and write, two-thirds 374 (66.4%) were living in urban residences, and 191 (33.9%) were housewives (Table 1).

### Clinical characteristics of adult HTN patients'

More than half 349 (62%) of the study participants were less than or equal to 5 years since the first HTN was diagnosed. Nearly half of 253 (44.9%) were overweight (BMI ≥ 25), and more than two-thirds of 402 (71.4%) were WHO stage II HTN (Table 2).

### Adult HTN patients' psychosocial and behavioral characteristics

More than half 348 (61.8%) of the study participants had depression. Regarding behavioral practices, 49 (8.7%) were currently chewing Khat, 94 (16.7%) had drunk alcohol, and almost all 561 (99.6%) of the study participants were never smokers (Table 3).

### Prevalence of poor sleep quality

The prevalence of sleep quality among adult HTN patients was found to be 212 (37.7%) with (95% CI; 33.4–41.6).

### Adult HTN patients' sleep characteristics

The study participants went to bed at an average of 9:36 pm and woke up in the morning at 5:08 am. Each night, the average time spent sleeping was 12:50 hrs (SD 1.68 hours). Of the total participants, 331 (58.3%) of HTN patients had very good subjective sleep quality, and 310 (55.1%) of them were sleeping for 16–30 minutes. Also, 101 (17.9%) of the study participants were sleeping < 5:00 per 24 hours (Table 4).

### Reasons for difficulty sleeping in adult HTN patients

Among the total participants, 231 (41%) of HTN patients were facing difficulty maintaining sleep within 30 minutes, followed by pain of 49 (8.7%) and cough of 35 (6.2%) (Fig 1).

### Factors associated with sleep quality

In the multivariable logistic regression analysis, female gender, age ≥ 65 years, being overweight (BMI ≥ 25 kg/m2), WHO stage II HTN patients, physical inactivity, depression, and anxiety were significantly associated with poor sleep quality.

**Table 1. Socio-demographic characteristics of adult HTN patients attending chronic follow-up care clinics in northwest Amhara regional state referral hospitals in Northwest Ethiopia in 2021 (n = 563).**

| Variables | Category | Frequency (no.) | Percent (%) |
|---|---|---|---|
| Sex | Male | 232 | 41.2 |
| | Female | 331 | 58.8 |
| Age group | 18–40 years | 87 | 15.5 |
| | 41–65 years | 208 | 36.9 |
| | ≥ 65 years | 268 | 47.6 |
| Ethnicity | Amhara | 504 | 89.5 |
| | Oromo | 21 | 3.7 |
| | Tigre | 31 | 5.5 |
| | Other | 7 | 1.2 |
| Marital status | Single | 35 | 6.2 |
| | Married | 347 | 61.6 |
| | Divorced | 47 | 8.3 |
| | Widowed | 134 | 23.8 |
| Religion | Orthodox | 435 | 77.3 |
| | Muslim | 93 | 16.5 |
| | Protestant | 30 | 5.3 |
| | Catholic | 5 | 0.9 |
| Educational status | Unable to read and write | 215 | 38.2 |
| | Primary school (1–8) | 120 | 21.3 |
| | Secondary school (9–12) | 87 | 15.5 |
| | College and above | 141 | 25 |
| Occupation | Government employee | 132 | 23.4 |
| | Daily labor | 54 | 9.6 |
| | Farmer | 112 | 19.9 |
| | Housewife | 171 | 30.4 |
| | Merchant | 79 | 14 |
| | Other | 15 | 2.7 |
| Residence | Urban | 374 | 66.4 |
| | Rural | 189 | 36. |
| Average monthly income | ≤ 1000 birr | 213 | 37.8 |
| | 1001–2000 birr | 110 | 19.5 |
| | 2001–3500 birr | 81 | 14.4 |
| | >3500 birr | 159 | 28.2 |

**Notes:** 'Other' indicates: Occupation includes retirement, driver and ethnicity includes kimant, and agew.

The odds of developing poor sleep quality among female HTN patients were 2.55 times higher than male HTN patients [AOR = 2.55, 95%CI (1.55–4.18)], The odds of developing poor sleep quality among the age group ≥ 65 years old adult HTN patients were 4.07 times higher than the age group 18–40 years old adult HTN patients [AOR = 4.07, 95% CI (2.08–7.97)], the odds of developing poor sleep quality were 1.68 times higher among overweight (BMI ≥ 25 kg/m2) HTN patients than patients weight (BMI = 18.5–24.9 kg/m2) [AOR = 1.68, 95% CI (1.06–2.65)], the odds of developing poor sleep quality were 1.78 times higher among HTN patients WHO stage II as compared with HTN patients WHO stage I [AOR = 1.78, 95% CI (1.01–3.12)], physically inactive HTN patients were 2.39 times more likely had poor sleep quality than patients who did regular physical activity [AOR = 2.39, 95% CI (1.40–4.05)], the

**Table 2. Clinical characteristics of adult HTN patients at chronic follow-up care clinics in northwest Amhara regional state referral hospitals, Northwest Ethiopia, 2021 (n = 563).**

| Variables | Category | Frequency (no.) | Percent (%) |
|---|---|---|---|
| Duration since first Dx of HTN | ≤ 5years | 349 | 61.9 |
| | 6–10 | 142 | 25.2 |
| | ≥ 11years | 72 | 12.8 |
| Duration of medication use | ≤ 5years | 372 | 66.1 |
| | 6–10 | 123 | 21.8 |
| | ≥ 11years | 68 | 12.1 |
| BMI | 18.5–24.9kg/m2 | 258 | 45.8 |
| | <18.5kg/m2 | 52 | 9.2 |
| | ≥ 25kg/m2 | 253 | 44.9 |
| Recent WHO stage of HTN | stage I | 125 | 22.2 |
| | Stage II | 402 | 71.4 |
| | Stage III | 24 | 4.3 |
| | Stage IV | 12 | 2.1 |
| Types of HTN | Systolic HTN | 127 | 22.6 |
| | Diastolic HTN | 122 | 21.7 |
| | Both types of HTN | 314 | 55.8 |
| Comorbidities | Yes | 158 | 28.1 |
| | No | 415 | 73.7 |
| Types of anti-HTN medication | Diuretics | 152 | 27 |
| | CCB | 133 | 23.6 |
| | ACIE | 278 | 49.4 |

**Notes:** ACIE: Angiotensive converting inhibitor enzyme, BMI: Body mass Index, CCB: Calcium channel blocker.

odds of poor sleep quality were 2.03 times developed depression than HTN patients who were free of depression [AOR = 2.03, 95% CI (1.23–3.44)],and HTN patients who had anxiety were 1.89 times more likely exposed to poor sleep quality than HTN patients free of anxiety [AOR = 1.89, 95% CI (1.16–3.03)] (Table 5).

**Table 3. Adult HTN patients' psychosocial and behavioral characteristics at chronic follow-up care clinics in northwest Amhara regional state referral hospitals, Northwest Ethiopia, 2021 (n = 563).**

| Variables | Category | Frequency (no.) | Percent (%) |
|---|---|---|---|
| Depression | Yes | 348 | 61.8 |
| | No | 215 | 28.2 |
| Anxiety | Yes | 183 | 32.5 |
| | No | 380 | 67.5 |
| Physical activity | Yes | 178 | 31.7 |
| | No | 385 | 68.4 |
| Drinking status | Past drinker | 94 | 16.7 |
| | Current drinker | 35 | 6.2 |
| | Never drinker | 434 | 77 |
| Smoking status | Former smoker | 2 | 0.4 |
| | Never smoker | 561 | 99.6 |
| Chewing status | Past chewer | 27 | 4.8 |
| | Current chewer | 49 | 8.7 |
| | Never chewer | 487 | 86.5 |

**Table 4. Adult HTN patients' sleep characteristics at chronic follow-up care clinics in northwest Amhara regional state referral hospitals, Northwest Ethiopia, 2021 (n = 563).**

| Variables | Category | Frequency (no.) | Percent (%) |
|---|---|---|---|
| Subjective sleep quality | Very good | 331 | 58.3 |
| | Fairly good | 90 | 16 |
| | Fairly bad | 56 | 9.1 |
| | Very bad | 91 | 16.1 |
| Sleep latency | < 15 minutes +not during the past month | 94 | 16.7 |
| | 16-30munite+once or twice a week | 310 | 55.1 |
| | 31-60minute+ twice a week | 146 | 25.9 |
| | >60 minute+ three times a week | 13 | 2.3 |
| Sleep duration | <7 hours | 316 | 56.1 |
| | 6–7 hours | 73 | 13 |
| | 5–6 hours | 75 | 14 |
| | < 5 hours | 101 | 17.9 |
| Habitual sleep efficiency | ≥ 85% | 348 | 61.8 |
| | 75–84% | 101 | 17.9 |
| | 65–74% | 55 | 9.8 |
| | <65% | 59 | 10.5 |
| Sleep disturbance | None | 328 | 58.2 |
| | 1–9 | 217 | 38.2 |
| | 10–18 | 7 | 1.2 |
| | 19–27 | 11 | 1.9 |
| Use sleep medication | Not during the past month | 513 | 91.1 |
| | Less than once a week | 22 | 3.9 |
| | once or twice a week | 15 | 2.7 |
| | Three or more times a week | 13 | 2.3 |
| Daytime dysfunction | no problem | 230 | 40.9 |
| | Slight problem | 239 | 42.5 |
| | Moderate problem | 86 | 15.3 |

## Discussion

In this study, the overall prevalence of poor sleep quality among adult HTN patients was 37.7% [95% CI: 33.4–41.6)]. The finding was in line with the studies done in Indonesia 35.6% [39], South Asia 34–41% [40], Italy 38.2% [12], and Jimma, Ethiopia 35.5% [11]. On the other hand, it was lower than the study done in the US 52.1% [41], China 57.1% [9, 42–45], Brazil 55% [21], Turkey 63.3% [46], Nigeria 43.2% [10, 47]. The discrepancy could be attributed to study design, study setting, sampling method, data collection tool, and/or procedures.

Likewise, this study revealed that the odds of developing poor sleep quality were 2.55 times higher in female HTN patients than in male HTN patients. This is supported by the studies done in Italy (10), Turkey (27), and Australia [48]. Females may have been more stressed as a result of increased household responsibilities, according to one possible explanation (job stress). This could be the sole cause of poor sleep quality [49]. Another reason could be that females are more prone to anxiety, depression, and insomnia, which could exacerbate poor sleep quality [50]. Or during the premenopausal or menopausal period, the level of estrogen and progesterone has decreased. This results in a two-fold increase in the number of arousals after sleep this could result in poor sleep quality [51].

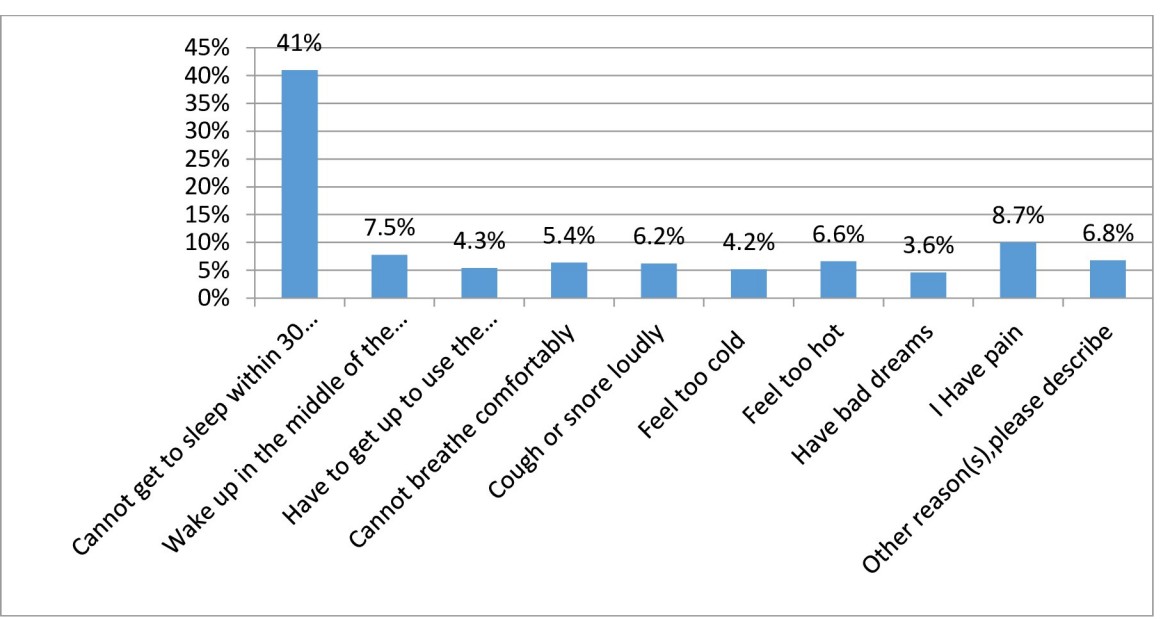

**Fig 1. Reasons for difficulty sleeping in adult HTN patients at chronic follow-up care clinics in northwest Amhara regional state referral hospitals in Northwest Ethiopia, 2021 (n = 563).**

The odds of developing poor sleep quality among the aged group $\geq 65$ years old adult HTN patients was 4.07 times higher than the age group of 18–40-years-old adult HTN patients. It is in agreement with the study done in Debre Markos, Ethiopia [13]. It implies that when age increases, poor sleep quality also increase [52].

Although the odds of developing poor sleep quality were 1.68 times higher among overweight (BMI $\geq 25$ kg/m2) HTN patients than among patients' weight (BMI = 18.5–24.9 kg/m2). This is consistent with the studies done in China [42] and Nigeria [53]. The possible justification could be that having excess body weight causes obstructive sleep apnea (difficulty of breathing during sleep), hence insufficient sleep due to breathing difficulties enhances poor sleep quality [54–57].

The odds of developing poor sleep quality were 1.78 times higher among HTN patients with WHO stage II as compared with HTN patients with WHO stage I. This study was supported by studies conducted in Nigeria [53] and Jimma, Ethiopia [11]. The possible reasons could be that high blood pressure makes patients have short sleep duration, low sleep efficiency, and insomnia [58–60].

Physically inactive HTN patients were 2.39 times more likely to have poor sleep quality than patients who did regular physical activity. This is supported by the studies conducted in Turkey [46] and Jimma, Ethiopia [11]. The possible justification might be that regular physical exercise improves sleep quality by decreasing obstructive sleep apnea severity [61, 62].

The odds of poor sleep quality were 2.03 times greater for those who had developed depression than for HTN patients who were free of depression, and hypertension patients who had anxiety were 1.89 times more likely to be exposed to poor sleep quality than HTN patients free of anxiety. This study was supported by studies conducted in Nigeria [53] and Australia [48]. The possible justification could be that during depression or anxiety, the level of serotonin and/or sleep-inducing chemicals is reduced. As a result, a patient might have a longer sleep onset latency, a greater number of arousals, greater wake-up time during the night, and fewer transitions into non-rapid eye movement (NREM) [17, 63, 64]. This might result in poor sleep quality.

**Table 5. Bivariable and Multivariable logistic regression analysis among adult HTN patients at chronic follow-up care clinic in northwest Amhara regional state referral hospitals, Northwest Ethiopia, 2021 (n = 563).**

|  | Sleep quality | | | |
|---|---|---|---|---|
| Variables | Poor | Good | COR (95%) | AOR (95%) |
| Sex |  |  |  |  |
| Male | 72 | 160 | 1 | 1 |
| Female | 140 | 191 | 1.63(1.14–2.31) | 2.547 (1.55–4.18)* |
| Age groups |  |  |  |  |
| 18–40 years | 28 | 59 | 1 | 1 |
| 41–65 years | 28 | 180 | 0.33 (0.18–0.59) |  |
| ≥ 65 years | 156 | 112 | 2.94 (1.76–4.89) | 4.07 (2.08–7.97)* |
| Marital status |  |  |  |  |
| Single | 7 | 28 | 1 |  |
| Married | 131 | 216 | 2.43 (1.03–5.71) |  |
| Divorced | 19 | 28 | 2.71 (0.99–7.47) |  |
| Widowed | 55 | 79 | 2.79 (1.14–6.83) |  |
| Educational status |  |  |  |  |
| Unable to read and write | 73 | 141 | 0.83 (0.54–1.29) |  |
| Primary school | 55 | 65 | 1.36 (0.83–2.24) |  |
| Secondary school | 30 | 58 | 0.83 (0.48–1.45) |  |
| College and above | 54 | 87 | 1 |  |
| Average monthly income |  |  |  |  |
| ≤ 1000 birr | 86 | 127 | 0.95 (0.63–1.45) |  |
| 1001–2000 birr | 35 | 75 | 0.66 (0.39–1.09) |  |
| 2001–3500 birr | 25 | 56 | 0.63 (0.36–1.11) |  |
| >3500 birr | 66 | 93 | 1 |  |
| BMI |  |  |  |  |
| 18.5–24.9 kg/m2 | 80 | 178 | 1 |  |
| <18.5 kg/m2 | 18 | 34 | 1.18 (0.63–2.21) |  |
| ≥ 25 kg/m2 | 114 | 139 | 1.83 (1.27–2.62) | 1.68 (1.06–2.65)* |
| Duration since first Dx of HTN |  |  |  |  |
| h≤ 5 years | 120 | 229 | 1 |  |
| 6–10 years | 64 | 78 | 1.57 (1.05–2.33) |  |
| ≥ 11 years | 28 | 44 | 1.21 (0.72–2.05) |  |
| Duration of anti-HTN medication use |  |  |  |  |
| ≤ 5 years | 127 | 245 | 1 |  |
| 6–10 years | 56 | 67 | 1.61 (1.07–2.44) |  |
| ≥ 11 years | 29 | 39 | 1.43 (0.85–2.43) |  |
| Recent WHO stage of HTN |  |  |  |  |
| Stage I | 30 | 95 | 1 |  |
| Stage II | 164 | 238 | 2.18 (1.38–3.44) | 1.78 (1.01–3.12)* |
| Stage III | 12 | 12 | 3.17 (1.29–7.78) |  |
| Stage IV | 6 | 6 | 3.17(0.95–10.55) |  |
| Physical activity |  |  |  |  |
| Yes | 38 | 140 | 1 | 1 |
| No | 174 | 211 | 3.04 (2.02–4.58) | 2.39 (1.40–4.05)* |
| Depression |  |  |  |  |
| Yes | 153 | 195 | 2.08 (1.44–2.99) | 2.03 (1.23–3.34)* |
| No | 59 | 156 | 1 | 1 |

(*Continued*)

**Table 5.** (Continued)

| | Sleep quality | | | |
|---|---|---|---|---|
| Anxiety | | | | |
| Yes | 96 | 87 | 2.51(1.75–3.61) | 1.89 (1.16–3.03)* |
| No | 116 | 264 | 1 | |

Notes: *indicates P-Value < 0.05.

## Conclusions

In this study, more than one-third of the study participants had poor sleep quality. Female gender, age $\geq$ 65 years, overweight (BMI) $\geq$ 25 kg/m2, stage II hypertension, physical inactivity, depression, and anxiety were all associated with poor sleep quality. Managing depression and anxiety disorders, encouraging physical activity, and providing health education about weight loss are all necessary for HTN patients.

## Supporting information

**S1 Data.**
(SAV)

## Acknowledgments

The authors acknowledge Teda Health Science College, northwest Amhara regional state referral hospitals' directors, nurses, study participants, data collectors, and supervisors for their support and contributions to this study.

## Author Contributions

**Conceptualization:** Teshale Ayanaw, Mahlet Temesgen, Abere Woretaw Azagew, Yohannes Mulu Ferede.

**Data curation:** Teshale Ayanaw, Yohannes Mulu Ferede.

**Formal analysis:** Teshale Ayanaw, Mahlet Temesgen, Abere Woretaw Azagew, Yohannes Mulu Ferede.

**Funding acquisition:** Teshale Ayanaw, Mahlet Temesgen, Abere Woretaw Azagew, Yohannes Mulu Ferede.

**Investigation:** Teshale Ayanaw, Mahlet Temesgen, Abere Woretaw Azagew, Yohannes Mulu Ferede.

**Methodology:** Teshale Ayanaw, Mahlet Temesgen, Abere Woretaw Azagew, Yohannes Mulu Ferede.

**Project administration:** Teshale Ayanaw, Abere Woretaw Azagew, Yohannes Mulu Ferede.

**Resources:** Mahlet Temesgen, Yohannes Mulu Ferede.

**Software:** Teshale Ayanaw, Mahlet Temesgen, Yohannes Mulu Ferede.

**Supervision:** Mahlet Temesgen, Abere Woretaw Azagew, Yohannes Mulu Ferede.

**Validation:** Teshale Ayanaw, Mahlet Temesgen, Abere Woretaw Azagew, Yohannes Mulu Ferede.

**Visualization:** Teshale Ayanaw, Abere Woretaw Azagew, Yohannes Mulu Ferede.

**Writing – original draft:** Teshale Ayanaw, Abere Woretaw Azagew, Yohannes Mulu Ferede.

**Writing – review & editing:** Mahlet Temesgen, Abere Woretaw Azagew, Yohannes Mulu Ferede.

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
