## [Decision Letter · Decision Letter 0]

11 Apr 2022

PONE-D-22-06339Sleep quality and associated factors among adult hypertensive patients attending at chronic follow up care clinic in northwest Amhara regional state referral hospitals, Northwest EthiopiaPLOS ONE

Dear Dr. Ferede,

Thank you for submitting your manuscript to PLOS ONE. After careful consideration, we feel that it has merit but does not fully meet PLOS ONE’s publication criteria as it currently stands. Therefore, we invite you to submit a revised version of the manuscript that addresses the points raised during the review process.

We look forward to receiving your revised manuscript.

Kind regards,

Giannicola Iannella, M.D

Academic Editor

PLOS ONE

Journal Requirements:

Additional Editor Comments:

Dear authors please revise the manuscript according to the reviewer's comments.

Regards

Reviewers' comments:

Reviewer's Responses to Questions

**Comments to the Author**

1. Is the manuscript technically sound, and do the data support the conclusions?

Reviewer #1: Yes

Reviewer #2: Partly

2. Has the statistical analysis been performed appropriately and rigorously? 

Reviewer #1: Yes

Reviewer #2: Yes

3. Have the authors made all data underlying the findings in their manuscript fully available?

Reviewer #1: Yes

Reviewer #2: Yes

4. Is the manuscript presented in an intelligible fashion and written in standard English?

Reviewer #1: Yes

Reviewer #2: No

5. Review Comments to the Author

Reviewer #1: english editing should be performed.

Introduction

- OSAS severity and complete blood count parameters assessed as inflammatory markers were examined, it was identified that the NLR and PLR levels were not very determinant in predicting either OSAS or its severity, and among these parameters, WMR was more significant, and it was determinant in distinguishing severe OSAS. These markers in the diagnosis of OSAS, determination of its severity and its monitoring. please discuss and cite doi:10.26355/eurrev_202106_26044

- line 77, osas were correlated with a greater risk of neurovegetative pathologies and a reduced cognitive performance in both adults and children. An interesting state of the art found Seventy papers reported neurocognitive Performance improvement in OSA patients after CPAP therapy. Eighty percent of studies found improved executive functions such as verbal fluency or working memory, with partial neural recovery at long-term follow-up. One article compared the effect of MAD, CPAP treatment on cognitive disorders, reporting better improvement of CPAP and MAD than placebo in cognitive function. please discuss and cite doi:10.3390/bs11120180

Methods

- please apply strobe guidelines.

- please add a flow diagram explaining the study protocol

- protocol well written

Resutls

add the p value in all the tables were possible

Discussion

- among the different treatments currently available for osas have been shown in a variable way to influence the associated comorbidities, reducing systemic inflammation, helping to reduce weight, cholesterol levels or blood pressure. However, not only medical treatment is proposed as a valid alternative but surgery in selected cases can be considered. please cite doi:10.5664/jcsm.9564 and doi:10.1016/j.amjoto.2021.103197

- line 321, The incidence of obstructive sleep apnea syndrome (OSAS) also in children has been investigated, increasing year over year recently. Research on OSAS children's blood pressure can help understand the onset of OSAS-related complications. Early diagnosis and intervention in blood pressure changes in children with OSAS can reduce the incidence of cardiovascular disease in late adulthood. Children with moderate to severe OSAS had higher blood pressure than those with PS and mild OSAS at all stages of sleep. Blood pressure in the REM sleep phase was higher than in the other sleep phases in all groups of children. please cite doi:10.1002/ped4.12163

Reviewer #2: The article presents an interesting topic considering the national origin of the research group. However, the introduction and discussion part appears to be exposed in a superficial and generic way. In addition, English language needs major revisions. For this reason I believe that the work can be taken into consideration for a possible publication by paying more attention to the linguistic part. The discussion part should be improved and better compared with the international literature.

6. PLOS authors have the option to publish the peer review history of their article (what does this mean?). If published, this will include your full peer review and any attached files.

Reviewer #1: No

Reviewer #2: **Yes: **Irene Claudia Visconti

---

## [Author Response · Author response to Decision Letter 0]

6 May 2022

Authors’ point-by-point Response to the editor and Reviewers Reports

Sleep quality and associated factors among adult hypertensive patients attending a chronic follow up care clinic in northwest Amhara regional state referral hospitals, Northwest Ethiopia.

Authors:

 Teshale Ayanaw1, Mahlet Temesgen2, Abere Woretaw Azagew1, Yohannes Mulu Ferede1*

First, we authors would like to give our heartfelt gratitude to the "PLOS ONE" Editorial Team and the respective reviewers for reviewing our manuscript and providing the necessary comments that are going to be corrected accordingly. As per the reviewers’ comments, the authors tried to make an amendment and corrections accordingly. This is a point-by-point letter to reviewer 1 and 2 reports. 

1. Reviwer-1 comments and author’ responses

Reviewer comment-1

English editing should be performed.

Authors’ response-1

Dear Reviewer, Thank you for your information. We tried to edit the entire document, including the English language. We tried to edit the language by using different online English editing services. 

Reviewer comment-2

Introduction

- OSAS severity and complete blood count parameters assessed as inflammatory markers were examined, it was identified that the NLR and PLR levels were not very determinant in predicting either OSAS or its severity, and among these parameters, WMR was more significant, and it was determinant in distinguishing severe OSAS. These markers in the diagnosis of OSAS, determination of its severity and its monitoring. please discuss and cite doi:10.26355/eurrev_202106_26044

- line 77, osas were correlated with a greater risk of neurovegetative pathologies and a reduced cognitive performance in both adults and children. An interesting state of the art found Seventy papers reported neurocognitive Performance improvement in OSA patients after CPAP therapy. Eighty percent of studies found improved executive functions such as verbal fluency or working memory, with partial neural recovery at long-term follow-up. One article compared the effect of MAD, CPAP treatment on cognitive disorders, reporting better improvement of CPAP and MAD than placebo in cognitive function. please discuss and cite doi:10.3390/bs11120180

Authors’ response-2

Dear reviewer, Thank you for your unending assistance in searching for and adding important references. We attempted to discuss and cite the references based on your recommendation.

Look at the discussion and citation on lines 78–83 of the clean version "manuscript," "the introduction part."

Reviewer comment-3

Methods

- please apply strobe guidelines.

- please add a flow diagram explaining the study protocol

- protocol well written

Authors’ response-3

Dear Reviewer, Thank you for your interesting idea.

• We used a strobe guideline.

• In terms of the flow diagram, it is unusual to publish it alongside the manuscript, but we explain it in detail in the sampling procedure/technique. We double-checked the sampling procedure and attempted to clarify how the samples are taken. Take a look at the flow diagram below.

• Flow diagram: “Among the total of five hospitals found in the Northwest Amhara region, three of them were selected by using the lottery method. And then samples were proportionally allocated in each selected hospital. Hereafter, the desired sample size was selected by using a systematic random sampling technique (every four HTN patients were selected)”. 

• Check line no 140-144 clean version “manuscript”

Reviewer comment-4

Results

add the p value in all the tables were possible

Authors’ response-4

Thanks for your advice: we put the p-value in the foot note”. *indicates P- Value <0.05. That is the way we omit the p-value from each variable.

Reviewer comment-5

Discussion

- among the different treatments currently available for osas have been shown in a variable way to influence the associated comorbidities, reducing systemic inflammation, helping to reduce weight, cholesterol levels or blood pressure. However, not only medical treatment is proposed as a valid alternative but surgery in selected cases can be considered. please cite doi:10.5664/jcsm.9564 and doi:10.1016/j.amjoto.2021.103197

- line 321, The incidence of obstructive sleep apnea syndrome (OSAS) also in children has been investigated, increasing year over year recently. Research on OSAS children's blood pressure can help understand the onset of OSAS-related complications. Early diagnosis and intervention in blood pressure changes in children with OSAS can reduce the incidence of cardiovascular disease in late adulthood. Children with moderate to severe OSAS had higher blood pressure than those with PS and mild OSAS at all stages of sleep. Blood pressure in the REM sleep phase was higher than in the other sleep phases in all groups of children. please cite doi:10.1002/ped4.12163

Authors’ response-5

Dear Reviewer, Again, we thank you for providing these important references. Based on your recommendation, we tried to cite the reference. 

Look at line number 321 (ref. no 58, doi:10.5664/jcsm.9564) & 326 (ref.no 61, doi:10.1002/ped4.12163)

2. Reviwer-2 comments and author’ responses

Reviewer comment-1

Reviewer #2: The article presents an interesting topic considering the national origin of the research group. However, the introduction and discussion part appears to be exposed in a superficial and generic way. In addition, English language needs major revisions. For this reason I believe that the work can be taken into consideration for a possible publication by paying more attention to the linguistic part. The discussion part should be improved and better compared with the international literature.

Authors’ response-1

Thanks dear, for reviewing and providing us with useful comments,

Based on your comments, we attempted to improve the English language by utilizing various online English editing services, such as online grammar checkers and QuillBot etc.

Similarly as much as possible we tried to revise and amend the entire document including the discussion part.

3. For editor

According to your inquiry, we attempted to improve the English language by using an online grammar checker and a QuillBot grammar checker, as well as revising and editing the entire manuscript.

---

## [Editor Report · Decision Letter 1]

23 Jun 2022

Sleep quality and associated factors among adult hypertensive patients attending a chronic follow up care clinic in northwest Amhara regional state referral hospitals, Northwest Ethiopia

PONE-D-22-06339R1

Dear Dr. Ferede,

We’re pleased to inform you that your manuscript has been judged scientifically suitable for publication and will be formally accepted for publication once it meets all outstanding technical requirements.

Kind regards,

Giannicola Iannella, M.D

Academic Editor

PLOS ONE

Additional Editor Comments (optional):

Very interesting paper
---

## [Editor Report · Acceptance letter]

27 Jun 2022

PONE-D-22-06339R1 

Sleep quality and associated factors among adult hypertensive patients attending a chronic follow up care clinic in northwest Amhara regional state referral hospitals, Northwest Ethiopia 

Dear Dr. Ferede:

I'm pleased to inform you that your manuscript has been deemed suitable for publication in PLOS ONE. Congratulations! Your manuscript is now with our production department. 

Kind regards, 

on behalf of

Dr. Giannicola Iannella 

Academic Editor

PLOS ONE